# Deciphering the Metabolic Pathways of Pitaya Peel after Postharvest Red Light Irradiation

**DOI:** 10.3390/metabo10030108

**Published:** 2020-03-14

**Authors:** Qixian Wu, Huijun Gao, Zhengke Zhang, Taotao Li, Hongxia Qu, Yueming Jiang, Ze Yun

**Affiliations:** 1Center of Economic Botany, Core Botanical Gardens, South China Botanical Garden, Chinese Academy of Sciences, Guangzhou 510650, China; wuqixian@scbg.ac.cn (Q.W.); wubangcai88@163.com (T.L.); q-hxia@scbg.ac.cn (H.Q.); ymjiang@scbg.ac.cn (Y.J.); 2Institute of Fruit Tree Research, Guangdong Academy of Agricultural Sciences, Guangzhou 510600, China; huijun_gao@aliyun.com; 3College of Food Science and Technology, Hainan University, Haikou 570228, China; zhangzhengke@hotmail.com

**Keywords:** fruit decay, red light, pitaya, primary metabolites, ROS-related characters, volatile compounds

## Abstract

Red light irradiation can effectively prolong the shelf-life of many fruit. However, little is known about red light-induced metabolite and enzyme activities. In this study, pitaya fruit was treated with 100 Lux red light for 24 h. Red light irradiation significantly attenuated the variation trend of senescence traits, such as the decrease of total soluble solid (TSS) and TSS/acidity (titratable acidity, TA) ratio, the increase of TA, and respiratory rate. In addition, the reactive oxygen species (ROS) related characters, primary metabolites profiling, and volatile compounds profiling were determined. A total of 71 primary metabolites and 67 volatile compounds were detected and successfully identified by using gas chromatography mass spectrometry (GC-MS). Red light irradiation enhanced glycolysis, tricarboxylic acid (TCA) cycle, aldehydes metabolism, and antioxidant enzymes activities at early stage of postharvest storage, leading to the reduction of H_2_O_2_, soluble sugars, organic acids, and C-6 and C-7 aldehydes. At a later stage of postharvest storage, a larger number of resistance-related metabolites and enzyme activities were induced in red light-treated pitaya peel, such as superoxide dismutase (SOD), ascorbate peroxidase (APX), 1,1-diphenyl-2-picryl-hydrazyl (DPPH) radical-scavenging, reducing power, fatty acids, and volatile aroma.

## 1. Introduction

Pitaya (*Hylocereus polyrhizus*) is a fruit crop of Latin America origin [1], and red pitaya is now widely cultivated in many other tropical and subtropical regions due to its attractive pulp color and high nutritional benefits [2]. Pitaya fruit contains a large amount of polyphenols, flavonoids, betalains, sugars, organic acids, amino acids, and phytalbumin [3], and consumers pay more attention to its beneficial antioxidant activity [4]. However, due to the perishability of pitaya fruit, its shelf life after harvest is only 4–6 days, seriously restricting the development of the pitaya fruit industry [5,6]. Over the past decade, researchers have developed a number of postharvest treatments that can effectively slow fruit decay. Apple polyphenols treatment delayed the browning and softening of fresh-cut red pitaya fruit, and maintained the contents of betacyanin and total phenolics in the fresh-cut pitaya fruit [1]. Ozone treatment significantly reduced the decay rate and maintained the total soluble solid (TSS) and TSS/titratable acidity (TA) ratio of pitaya fruit [5]. Green tea extract could effectively delay the growth of pathogens on fresh-cut pitaya fruit and prolong its shelf-life [2]. Most of these treatments are chemical treatments, which has a certain impact on human health. In addition, the production cost of extract or the processing cost of chemical treatment is also an important aspect that affects its application promotion. Therefore, the developing green, safe and low-cost preservation treatment has high application value.

Light is not only the energy source for plants, but also an essential signal for plant growth and development [7]. Red light could enhance photosynthetic rate of strawberry [8] and function as a global signal in *Pseudomonas syringae pv. syringae* strain B728a [9]. In addition, red light is an effective postharvest treatment, which has also been applied in the preservation of many kinds of fruits, such as accelerating the ripening of green tomato fruit [10], extending the storage period for broccoli [11], promoting the color transformation of fruit peel of harvest mandarins [12], and increasing anthocyanin accumulation in grape berry [13]. However, the effects of red light irradiation on pitaya fruit is not clear after harvest.

Primary metabolites and volatile compounds are main components of fruit quality and flavor, and the characteristic flavor metabolites in different fruit are different [5,14,15,16,17]. Fruit quality and flavor-related metabolites were significantly influenced by many endogenous and exogenous factors, such as natural ripening [14], natural senescence [17], storage conditions [18], postharvest treatments [6,15,19]. In recent years, the development of metabolomics technology provides a new way to reveal the physiological complexity of fruit. Many scholars have preliminarily analyzed many biological phenomena by method of metabolomics, such as evolutionarily distinct in natural variation of rice [20], the appearance/taste-oriented tomato breeding [21], fruit aroma components, and their changes during development and storage [22]. However, little is known about the metabolites induced by red light.

In this study, we first evaluated the effect of red light irradiation on fruit quality and shelf-life of pitaya. Subsequently, reactive oxygen species (ROS)-related characters, primary metabolites, and volatile compounds of pitaya peel were analyzed. We hypothesized that red light irradiation may have an effect on the peel metabolism of pitaya, directly or indirectly increasing resistance to fruit decay. ROS-related characters and metabolites that play a possible role in red light-induced resistance will be highlighted in comparison with fruit senescence.

## 2. Results

### 2.1. The Effects of Red Light Irradiation on Pitaya Fruit 

In this study, pitaya fruit was harvest at mature stage and irradiated for 24 h with 100 Lux red light emitting diode (LED) light. After ten days of postharvest storage, the fruit decay of pitaya was significantly delayed after red light irradiation (Figure 1a,b). The decay rate reduced from 56.21% in the control fruit to 12.53% in the red light-treated fruit (Figure 1c), indicating that red light irradiation can effectively delay the decay of pitaya fruit.

To study the effect of red light irradiation on fruit quality of pitaya, respiratory rate, TSS and TA were determined in this study. The respiratory rate of control pitaya increased significantly during the postharvest storage, while the respiratory rate of red light-treat fruit maintained on Day 0 level until the later stage of storage (Figure 1d). During postharvest storage, the TA content also increased significantly in control fruit, and red light irradiation effectively delayed the increase of TA content (Figure 1f). In contrast, the TSS content of control fruit decreased gradually from 0 d to 7 d, while the TSS content of red light-treated fruit remained at a higher level (Figure 1e). In addition, the decrease of TSS–TA ratio of pitaya fruit was significantly delayed by red light irradiation during postharvest storage (Figure 1g). This indicated that red light irradiation not only significantly delayed the decay of pitaya fruit, but also effectively maintained the fruit quality of pitaya.

### 2.2. ROS-Related Characters of Pitaya Peel

In order to evaluate the effect of red light irradiation on the ROS levels, we determined the content of H_2_O_2_ and ·OH in the pitaya peel. The content of H_2_O_2_ of control peel gradually increased during the postharvest storage, while the content of H_2_O_2_ decreased significantly after red light irradiation and maintained on Day 1 level until later stage of storage (Figure 2). During the postharvest storage, there was no significant change in ·OH content in both control and red light-treated peel (Figure 2). It is suggested that H_2_O_2_ may play a negative role in red light delaying the decay of pitaya fruit. 

In this study, the enzyme activity of four reactive oxygen scavengers were measured, including peroxidase (POD), ascorbate peroxidase (APX), catalase (CAT), and superoxide dismutase (SOD). Result showed that the activities of four enzymes varied with three trends. Firstly, the POD activity of red light-treated peel increased significantly on Day 1 and decreased significantly at 4 and 7 d compared with the control peel (Figure 2). Secondly, the activities of APX and SOD of red light-treated peel increased significantly from 1 d to 7 d compared with control peel (Figure 2). Thirdly, the activity of CAT of red light-treated peel decreased significantly on Day 1–7 compared with control peel (Figure 2), which was consistent with the variation trend of H_2_O_2_ content.

In addition, we measured both DPPH radical-scavenging activity and reducing power, which decreased gradually in control peel during postharvest storage (Figure 2). After red light irradiation, both DPPH free radical scavenging activity and reducing power significantly decreased on Day 1, but significantly increased at the later stage of postharvest storage (Figure 2). We speculated that DPPH free radical scavenging activity and reducing power might play a vital role in red light irradiation improving fruit resistance at later stage of postharvest storage.

In sum, the postharvest storage of red light treated fruit can be divided into two main stages according the variation trends of fruit quality characters and ROS-related characters. The first stage is 0–1 d, also known as the enhanced resistance stage, during which the activities of antioxidant enzymes was significantly increased. The second stage is 1–7 d, also known as the senescence stage, during which fruit quality and antioxidant capacity maintained at a high level compared with the control fruit.

### 2.3. The Changes of Primary Metabolites after Red Light Irradiation

In order to analyze the effect of red light irradiation on pitaya metabolites, the primary metabolite profiling of pitaya peel were analyzed by using gas chromatography mass spectrometry (GC-MS). Since 1 d and 7 d are typical time points for red light-treated pitaya fruit, primary metabolites of 1 d and 7 d samples were determined in this study. Among the 71 metabolites detected in pitaya peel, 55 were significant changed (*p* < 0.05) after red light irradiation, mainly including soluble sugars, amino acids, alcohols, organic acids, and fatty acids. 

To visualize the general clustering trends, projection to latent structures discriminant analysis (PLS-DA) were applied to the significantly changed primary metabolites obtained from the control and red light-treated pitaya peel. To assess the risk that the current PLS-DA model was spurious, the permutation test for PLS-DA was analyzed. All regression coefficient (R2) and cross-validated R2 (Q2) values on the left were lower than the original points on the right (Figure 3a), indicating that the PLS-DA model was valid. The samples of the control group and the red light irradiation group on day 1 and day 7 were separated clearly according to variable t[1] in Figure 3b. According to t[2], control and red light-treated samples were further apart on Day 7 and close to each other on Day 1 (Figure 3b). The result of hierarchical clustering analysis (HCA) further confirmed that the effect of red light irradiation on primary metabolites of pitaya peel was less than that of its own senescence (Figure 3c). To further analysis of the key primary metabolites in pitaya peel, orthogonal projection to latent structures discriminant analysis (OPLS-DA) was analyzed between two random groups of samples. A total of 22 primary metabolites were considered to be key metabolites (variable importance in projection (VIP) > 1) and were marked in red circles in Figure 3d. 

#### 2.3.1. Soluble Sugars Changes after Red Light Irradiation

Soluble sugar is the basic metabolite and key component to maintain normal energy and carbon supply during fruit life. In this study, we detected and successfully identified fourteen soluble sugars. After the analysis of ANOVA and OPLS-DA, seven soluble sugars were considered as key soluble sugars with VIP > 1 and *p* < 0.05, including D-fructose, D-glucose, mannose, sorbose, D-turanose, glucopyranose, and D-glucopyranoside (Figure 4). The D-glucose content of pitaya peel was the highest in soluble sugars, and glucopyranose and D-glucopyranoside are derivatives of glucose. During the postharvest storage of control fruit, the mannose content increased significantly, D-glucopyranoside and D-turanose contents decreased significantly, and the contents of the other four soluble sugars did not change significantly (Figure 4). After red light irradiation, the contents of all seven soluble sugars in red light-treated samples were significantly than those of the control samples, and the decrease continued until the later stage of storage (Figure 4). The fructose, glucose, mannose, and sorbose are important substrate for the glycolysis. It seemed that red light irradiation accelerated the glycolysis, and the consumption was not recoverable.

#### 2.3.2. Changes of Cell Wall Degradation Products after Red Light Irradiation

Fruit senescence is often accompanied by fruit softening and cell wall degradation. In this study, a total of ten cell wall degradation products were detected and identified in pitaya peel. After the analysis of ANOVA and OPLS-DA, six cell wall degradation products were considered as key soluble sugars with VIP > 1 and *p* < 0.05, including galactaric acid, D-gluconic acid, galactonic acid, galacturonic acid, glucaric acid, and myo-inositol (Figure 4). Galactaric acid and galacturonic acid are derivatives of galactonic acid, and glucaric acid is a derivative of gluconic acid. During the postharvest storage of control fruit, the contents of D-gluconic acid, galactonic acid, and galacturonic acid increased significantly (Figure 4), indicating that those three cell wall degradation products play an important role in the senescence of pitaya peel. After red light irradiation, the contents of all six cell wall degradation products decreased significantly on Day 7, while only three of them decreased on Day 1 (Figure 4). Strangely, the myo-inositol content decreased during the postharvest storage of both control and red light-treated fruit. We speculated that red light irradiation had little effect on the cell wall degradation of pitaya fruit at the enhanced resistance stage, but had a prominent delaying effect at the senescence stage.

#### 2.3.3. Fatty Acids Changes after Red Light Irradiation

In this study, we detected twelve fatty acids in pitaya peel, including five short-chain fatty acids and seven very-long-chain fatty acids. After the analysis of ANOVA and OPLS-DA, four very-long-chain fatty acids were considered as key metabolites with *p* < 0.05 and VIP > 1, including octadecanoic acid, hexadecanoic acid, eicosanoic acid, and docosanoic acid (Figure 4). During the postharvest storage of control fruit, the contents of hexadecanoic acid, eicosanoic acid and docosanoic acid decreased significantly, while the octadecanoic acid content increased significantly (Figure 4). Compared with the control peel, the contents of all four key fatty acids decreased on Day 1 after red light irradiation, but significantly increased on Day 7 after red light irradiation, except docosanoic acid (Figure 4). The results showed that red light irradiation accelerated consumption of very-long-chain fatty acids at the enhanced resistance stage and promoted the accumulation of very-long-chain fatty acids at the senescence stage. We speculated that the effect of red light irradiation on fatty acids in pitaya peel was different in different stages of postharvest storage.

#### 2.3.4. Organic Acids and Amino Acids Changes after Red Light Irradiation

In this study, we detected and successfully identified fifteen organic acids and ten amino acids in pitaya peel by searching National Institute of Standards and Technologies (NIST) and NIST05 database. After the analysis of ANOVA and OPLS-DA, three organic acids and one amino acid were considered as key metabolites with *p* < 0.05 and VIP > 1, including malic acid, ethanedioic acid, pentanedioic acid and tyrosine (Figure 4). During the postharvest storage of control fruit, the contents of malic acid, pentanedioic acid, and tyrosine decreased significantly, while the ethanedioic acid content increased significantly (Figure 4). Compared with the control peel, the contents of malic acid and pentanedioic acid decreased significantly after red light irradiation on Day 1, while the contents of ethanedioic acid, pentanedioic acid, and tyrosine increased significantly on Day 7, except docosanoic acid (Figure 4). The malic acid, ethanedioic acid, and pentanedioic acid are important substrates for the tricarboxylic acid (TCA) cycle. We speculated that red light irradiation might accelerate the TCA cycle at the enhanced resistance stage and slow the TCA cycle at the senescence stage. 

### 2.4. The Changes of Volatile Compounds after Red Light Irradiation

Volatile compounds are the main components of fruit aroma. In this study, headspace solid phase microextraction (SPME) coupled with GC-MS method was used to analyze the volatile compounds profiling of pitaya peel, so as to evaluate the effect of red light irradiation on the pitaya aroma. Consistent with the determination of primary metabolites, only 1 d and 7 d samples were used for volatile compounds profiling detection in this study. We detected 67 volatile compounds in pitaya peel, among which 49 showed significant changes (*p* < 0.05) after red light irradiation, mainly including aldehydes, esters, alkanes, ketones, terpenes, and alcohols. 

Due to the different determination methods of volatile compounds and primary metabolites, we established the PLS-DA model respectively. In this study, PLS-DA were applied to visualize the general clustering trends of 49 significantly changed volatile compounds between control and red light-treated samples. In order to further verify the established model, we conducted 100-time permutation test for the corresponding model. The result showed that all the R2 and Q2 values in the permutation test were lower than the original ones (Figure 5a). The Y-axis intercept for Q2 was below zero (0.0, −0.451). Those results validate that the current PLS-DA model is highly reliable.

The control and red light-treated samples were clearly separated on both Day 1 and Day 7 according to t[1] in Figure 5b. Interestingly, the red light-treated peel samples on day 1 and day 7 were clearly separated, while the control peel samples on day 1 and day 7 were not (Figure 5b). The HCA result further confirmed that the volatile compounds changed greatly after red light irradiation (Figure 5c). It indicated that the effect of red light irradiation on the volatile compounds of the pitaya peel was much greater than that of pitaya peel senescence. To further analysis of the key volatile compounds in pitaya peel, OPLS-DA was analyzed between two random groups of samples. A total of eleven volatile compounds were considered to be key metabolites (VIP > 1) and marked in red circles in Figure 5d. 

#### 2.4.1. Aldehydes Changes after Red Light Irradiation

Aldehydes are a kind of important substances that constitute fruit aroma. In this study, we detected twelve aldehydes in pitaya peel by using GC-MS. After the analysis of ANOVA and OPLS-DA, four aldehydes were considered as key volatiles with *p* < 0.05 and VIP > 1, including hexanal, 2-hexenal, 2-heptenal, and 4-heptenal (Figure 6). Among the volatile compounds in pitaya peel, the content of hexanal is the highest, followed by 2-hexenal (Figure 5d). The contents of hexanal, 2-heptenal and 4-heptenal in control peel decreased significantly during the postharvest storage (Figure 6). Compared with the control peel, the contents of all four key aldehydes decreased significantly on both Day 1 and Day 7 after red light irradiation (Figure 6). It indicated that C-6 aldehydes were characteristic aroma of pitaya fruit, and red light irradiation could significantly reduce the C-6 and C-7 aldehydes in pitaya fruit.

#### 2.4.2. Alcohols Changes after Red Light Irradiation

Alcohols are also known as a kind of important components of fruit aroma. In this study, we detected and successfully identified five volatile alcohols in pitaya peel by searching NIST05 and NIST database. After the analysis of ANOVA and OPLS-DA, 1-hexanol and β-linalool were considered as key volatiles with *p* < 0.05 and VIP > 1 (Figure 6). The 1-hexanol content was the third highest in the volatile compounds of pitaya peel, only lower than 2-hexenal, and hexanal. During the postharvest storage of control peel, the 1-hexanol content increased, while the β-linalool content decreased (Figure 6). After red light irradiation, the 1-hexanol content increased significantly on both Day 1 and Day 7 compared with the control peel, while the β-linalool content increased significantly only on Day 7 (Figure 6). We speculated that red light irradiation could promote the conversion of C-6 aldehydes to 1-hexanol, and 1-hexanol and β-linalool might play a role in red light irradiation, delaying fruit senescence of pitaya.

#### 2.4.3. Ketones Changes after Red Light Irradiation

In this study, we detected ten ketones in pitaya peel. After the analysis of ANOVA and OPLS-DA, two ketones were considered as key volatiles with *p* < 0.05 and VIP > 1, including 2-hydroxy-cyclopentadecanone and cyclohexenone (Figure 6). During the postharvest storage of control fruit, the contents of both 2-hydroxy-cyclopentadecanone and cyclohexenone decreased significantly (Figure 6). After red light irradiation, the content of 2-hydroxy-cyclopentadecanone increased significantly on Day 7 compared with the control peel, while the content of cyclohexenone decreased significantly on Day 1 (Figure 6). We speculated that 2-hydroxy-cyclopentadecanone might play a vital role in red light irradiation delaying fruit senescence of pitaya, and red light irradiation accelerated the consumption of cyclohexenone.

#### 2.4.4. Changes of Other Volatile Compounds after Red Light Irradiation

In this study, we also detected and successfully identified other forty volatile compounds (except aldehydes, ketones alcohols) in pitaya peel by searching NIST05 and NIST database, including eleven esters, six terpenes, fourteen alkanes, and nine others. After the analysis of ANOVA and OPLS-DA, three were considered as key volatiles with *p* < 0.05 and VIP > 1, including 2-octenal, nitro m-xylene, and palmitoleic acid (Figure 6). During the postharvest storage of control fruit, the contents of all three key other volatiles decreased significantly (Figure 6). After red light irradiation, the content of 2-octenal and palmitoleic acid increased significantly on Day 7 compared with the control peel (Figure 6). We speculated that 2-octenal and palmitoleic acid might play a role in red light irradiation delaying fruit senescence of pitaya.

### 2.5. Coordinated Changes in Metabolites and Physiological Characters after Red Light Irradiation

To establish a link between key metabolites and red light irradiation effects, we analyzed the metabolites related to or co-regulated by the physiological characters of pitaya peel. Pairwise correlation analysis was used to evaluate the coordinated shift between key metabolites and physiological characters, and 33 key metabolites and twelve physiological characters were performed in this study. Clustering result showed that the decay rate was positively correlated with CAT,·OH, H_2_O_2_, respiratory rate and TA, but negatively correlated with POD, APX, SOD, TSS, DPPH radical-scavenging and reducing power (Figure 7). All physiological characters of pitaya could be divided into senescence traits and anti-senescence traits. After red light irradiation, the senescence traits were significantly reduced or delayed, and the anti-senescence traits were significantly enhanced or maintained (Figure 8). 

The key metabolites could be classified into four clusters according to their correlation with senescence traits and anti-senescence traits (Figure 7). In Cluster 1, six key metabolites were negatively correlated with most anti-senescence traits and positively correlated with most senescence traits, including three cell wall degraded products, mannose, octadecanoic acid, and ethanedioic acid (Figure 7). The content of the six key metabolites increased the most during the postharvest storage. After red light irradiation, the contents of only octadecanoic acid and ethanedioic acid increased on Day 7 compared with control peel, indicating that octadecanoic acid and ethanedioic acid might be related to anti-senescence.

In Cluster 2, all fifteen key metabolites were negatively correlated with the activities of APX and SOD, mainly including six soluble sugars and four volatile aldehydes (Figure 7). We speculated that there might be an interaction between the antioxidant enzymes and the consumption of soluble sugars and volatile aldehydes after red light irradiation (Figure 8). 

In Cluster 3, all four key volatile aroma were positively correlated with most senescence traits, including 1-hexanol, β-linalool, palmitoleic acid, and 2-hydroxy-cyclopentadecanone (Figure 7). The contents of all four key volatile aroma increased significantly on Day 7 after red light irradiation, indicating that those four volatile aroma might play a positive role in anti-senescence (Figure 8).

In Cluster 4, seven key metabolites were positively correlated with anti-senescence traits and negatively correlated with most senescence traits, mainly including eicosanoic acid, malic acid, pentanedioic acid, tyrosine, 2-octenal, nitro m-xylene, and myo-inositol (Figure 7). The contents of those seven key metabolites decreased significantly during the postharvest storage, indicating that the consumption of those metabolites might be related to the senescence of pitaya fruit.

## 3. Discussion 

The senescence is the final stage of fruit development and ripening. During the senescence, the fruit gradually deteriorated and disordered, which contribute to the weakening of peel structure and leads to the final demise of the fruit, such as peel thinned, water lose, fruit softening, TSS reduction, and respiratory rate increasing [14,15,18,23]. In addition, there are a variety of fungal pathogens on the fruit surface. As the senescence deepens, fungal pathogens are more likely to break through the fruit’s defenses and grow, spread and reproduce, accelerating the deterioration and disorder of fruit [24,25]. The fruit-pathogen interactions played a decisive role in accelerating the ripening and senescence of fruit and leading to the deterioration of fruit quality [24].

Previous studies have found that postharvest treatment can significantly slow down senescence as much as possible to preserve fruit quality [5,26,27]. In the very final stage of fruit development, this simple operation made great economic value to both growers and consumers. In this study, red light irradiation not only significantly delayed the senescence and decay of pitaya fruit, but also effectively maintained fruit quality of pitaya fruit; thus, prolonging the shelf-life of pitaya (Figure 1). Similarly, the blue light treatment significantly slowed the senescence and decay of pitaya fruit [6]. However, due to the long processing time of red light and the need for direct irradiation, no mature postharvest equipment has yet adopted red light for commercial processing, which requires subsequent development and research. Since the peel is the first barrier for pathogens to invade the fruit, delaying peel senescence improves fruit resistance and extends its shelf-life, which potentially enhances the crop value, and contribute to growers’ returns [24]. Therefore, the metabolic profiling analysis was employed to explore the details of the pitaya peel changes after red light irradiation in this study.

### 3.1. H_2_O_2_ Mediated the Peel Senescence of Pitaya 

Senescence in plants is a complex deterioration process that is regulated by autonomous (internal) factors and environmental signals, including age, reproductive development, and phytohormone levels, photoperiod, drought, cold, nutrient deficiency, wounding and shading [28]. By the time the fruit reaches harvest standards, its seeds have already developed and its value is to attract animals to spread the seeds. Unlike plant senescence, fruit senescence is a process of natural apoptosis, during which there is no nutrient reflux reuse and other links. Physiological activities during fruit senescence have been analyzed in previous studies, but the molecular triggers of this biological process are not yet clear.

ROS is one of the earliest responses of plant cells under various abiotic and biotic stresses and natural course of senescence [23]. Reactions involving ROS are an inherent feature of plant cells and contribute to a process of oxidative deterioration that may lead ultimately to cell death [23,29]. In this study, the content of H_2_O_2_ increased significantly during the postharvest storage of pitaya fruit, while the ·OH content changed slightly during this process (Figure 2). After red light irradiation, the content of H_2_O_2_ decreased significantly, and the fruit senescence and decay was significantly delayed (Figure 1 and Figure 2). It suggested that H_2_O_2_ might play the mediated role in the senescence of pitaya peel. Previous study reported that H_2_O_2_ was a cytotoxic molecule at high levels and a signaling molecule at low levels that regulates the gene expression associated with stress responses [30]. In this study, red light irradiation might not only significantly delay fruit senescence and decay of pitaya but also induce fruit resistance by reducing H_2_O_2_ content. On the contrary, blue light treatment significantly increased the H_2_O_2_ content in pitaya on Day 1 [6], indicating that blue light and red light irradiations had significant differences in the regulation of ROS signals in pitaya peel.

Lin et al. [31] reported that H_2_O_2_ treatment could accelerate pulp deterioration occurrence and shorten the shelf-life of longan fruit by enhancing the respiration rate. Cell wall degradation is an important aspect of fruit ripening and deterioration [14,18]. In this study, the content of cell wall degraded products increased significantly during the postharvest storage of pitaya peel. After red light irradiation, the increasing trend of cell wall degradation products was significantly delayed (Figure 4 and Figure 8), indicating that red light irradiation could delay the cell wall degradation of pitaya peel by reducing the content of H_2_O_2_. Similar results of H_2_O_2_ treatment promoting the degradation of fruit cell walls have also been reported in Kyoho grape [32], and the degradation of the fruit cell wall could be significantly delayed by reducing the content of H_2_O_2_ in pitaya fruit [5,6]. 

### 3.2. Red Light Irradiation Induced Fruit Resistance at Early Stage of Postharvest Storage

Plants have a complete system of enzymatic antioxidants to scavenge endogenous excess ROS [23]. SOD is very active in almost all physiological and biochemical processes in response to various stress in plants, and plays an important role in inducing plant stress resistance, such as resistance to environment stress, disease resistance, and anti-senescence [33]. PODs, well known as H_2_O_2_ oxidoreductases, are also involved in a broad range of physiological processes, such as lignin and suberin formation, cross-linking of cell wall components, phytoalexins synthesis, and starting the hypersensitive response (HR, a form of programmed host cell death at the infection site associated with limited pathogen development) [34]. Overexpression of APX in tobacco has been reported not only to reduce the toxicity of H_2_O_2_, but also to enhance tolerance to salt and water stress [35]. In this study, the activities of SOD, POD, and APX were induced by red light irradiation at the early stage of storage, indicating that SOD, POD, and APX played an important regulatory role in red light enhancing fruit resistance and delaying fruit senescence.

The glycolysis and TCA cycle are essential for energy provision in a wide range of physiological functions [36]. It is reported that plant glycolysis is induced by diverse biotic and abiotic stresses, such as pathogen attack, drought, flood, and hypoxia [37,38]. In this study, red light irradiation significantly enhanced glycolysis, and the key soluble sugar content significantly decreased after red light irradiation (Figure 4), indicating that red light irradiation could enhance the resistance of pitaya fruit by strengthening glycolysis at the early stage of storage. On the contrary, blue light treatment significantly reduced glycolysis rate in pitaya on Day 1 [6], indicating that blue light and red light irradiation had significant differences in glycolysis regulation in pitaya peel. Accelerating glycolysis and TCA cycle can provide more substances and energy for the defense against salt stress, thus improving the tolerance of cucumber seedlings to salt stress [39]. It seemed that TCA cycle might also play an important role in red light irradiation inducing the resistance of pitaya fruit. In this study, the substrates of TCA cycle were significantly reduced after red light irradiation at the early stage of storage (Figure 4). 

Previous studies have found that redox regulation of glycolysis and TCA cycle metabolic pathways seem to have a fundamental role in photosynthetic organisms [40]. Treatment of Arabidopsis thaliana with H_2_O_2_ caused the reduction in protein abundance of glycolysis and TCA cycle related enzymes [41]. The enzymes activities of glycolytic and TCA cycles were also reduced by oxidative post-translational modifications [42]. In this study, red light irradiation could remove H_2_O_2_ by increasing antioxidant enzyme activity; thus, enhancing glycolysis and TCA cycle of pitaya peel. However, the details of the interaction of antioxidant enzymes with glycolysis and TCA cycle need further analysis. 

The plant hormone jasmonic acid (JA) has been characterized as a central regulator of plant responses to abiotic and biotic stresses, and JA biosynthesis is catalyzed by lipoxygenases (LOX) [43]. Interestingly, overexpression of LOX gene in Arabidopsis resulted in marked reduction in volatile C-6 aldehydes [44]. Overexpression of aldehyde oxidase 4 in Silique can not only reduce the content of aldehydes but also significantly delay the senescence [45]. Although it has not been reported, we speculate that there seems to be some correlations between the reduction of aldehydes and the activation of JA signal. In addition, aldehydes offer the green-note aroma in many fruit, and their contents tend to decrease during fruit ripening and senescence [22]. C-6 and C-7 aldehydes are the characteristic aroma of pitaya peel [5,6]. After red light irradiation, the contents of C-6 and C-7 aldehydes significantly decreased (Figure 6), indicating that red light irradiation could improve the resistance of pitaya peel by inducing JA signal.

### 3.3. Red Light Irradiation Induced Fruit Resistance of Pitaya Peel at Later Stage of Storage

During the enhanced resistance stage, we found that the antioxidant enzymes played an important regulatory role in red light enhancing fruit resistance and delaying fruit senescence. At the senescence stage, the activity of SOD and APX in the red light-treated peel was significantly higher than that of the control, while the activity of POD was significantly lower than that of the control (Figure 2). It suggested that only SOD and APX played the regulatory role in red light enhancing fruit resistance and delaying fruit senescence at the senescence stage. Interestingly, the DPPH radical-scavenging activity and reducing power were induced by red light irradiation at the senescence stage. The DPPH radical-scavenging activity and reducing power are widely used for evaluating total antioxidant activity of low-molecular-mass metabolites, including phenolics, ascorbic acid, glutathione, tocopherols, and carotenoids [19]. We speculated that red light irradiation might increase the secondary metabolites in pitaya peel at the senescence stage; thus, enhancing the fruit resistance and delaying the fruit senescence. 

Fatty acids are the primary components of lipids, triacylglycerols and waxes/cutin which are essential not only as membrane constituents, but also are essential for growth, development, resistance to water loss and disease [46]. Plants utilize a combination of induced defense responses and preformed physical barriers like the epidermal wax, cuticle and cell wall to resist pathogens [44]. Very-long-chain fatty acids have strong water retention and can effectively inhibit plant water loss [47]. Overexpression of desaturase increased the accumulation of fatty acids in transgenic eggplants, and this was associated with increased resistance to Verticillium dahlia [48]. In addition, recent work has demonstrated that fatty acids may play an important role in systemic acquired resistance [44]. In this study, the key very-long-chain fatty acids were induced by red light irradiation at the senescence stage (Figure 4). We speculated that red light irradiation might induce the very-long-chain fatty acids in pitaya peel at the senescence stage, thus enhancing the fruit resistance and delaying the fruit senescence. On the contrary, blue light treatment significantly induced the fatty acids in pitaya peel on Day 1, indicating that blue light and red light irradiation had a time difference in the induction of fatty acid in pitaya peel.

Volatile aroma is an important index of fruit flavor and play an important role in responding to fruit senescence, temperature stress and plant disease [22]. It has been reported that the increase of terpenes at the later stage of storage has a positive effect in ozone treatment delaying the decay of citrus fruit [16]. The increase of volatile aroma also had a positive effect on melatonin prolonging the shelf-life of banana fruit [15]. In this study, the contents of β-linalool, 1-hexanol, 2-hydroxy-cyclopentadecanone, and palmitoleic acid increased by leaps and bounds after the red light irradiation at the senescence stage (Figure 6). Although the role of these volatile compounds in fruit senescence has not been reported so far, we speculate that they may be related to the resistance induced by red light irradiation at the senescence stage.

## 4. Materials and Methods

### 4.1. Plant Materials and Treatments

‘Hongshuijing’ pitaya fruit (*Hylocereus polyrhizus*) was harvested at the commercially mature stage (the sweetness of pulp increased significantly, the peel color changed from cyan to fuchsia) from a commercial orchard in Guangzhou City, China. After removing the scarred and diseased fruit, the fruit with uniform shape, color and size was selected for subsequent experiments. Fruit was continuously irradiated at 25 °C for 24 h under 100 Lux red light emitting diode (LED) light (660 nm), or in the dark (control). The distance from the LED to the fruit was 40 cm, and the fruit was flipped twice in equal time during the irradiation. After treatment, twelve fruit was packed into a plastic polyethylene box (48 × 30 × 13.5 cm) and then covered with one polythene plastic bag (0.03 mm, micro perforated). About 50 boxes for each treatment were stored in a room under 25 ± 2 °C and 75%–95% relative humidity. The fruit of each treatment was randomly taken out for peel sampling. The pulp and scales were removed and the equatorial part of the peel was taken for sampling. If part of the peel had decayed, the un-decayed portion was sampled. Twelve fruit for each biological replicate was sampled, and a total of five biological replicates were collected for each sample. Samples were taken at 0, 1, 4, and 7 d after harvest, and frozen in liquid nitrogen and stored at −80 °C for further analysis. 

### 4.2. Fruit Decay Rate Evaluation

After ten days of storage, the decay area on each peel of 50 fruit was calculated by the reported method [5,16] to evaluate the decay rate of pitaya fruit. Decay scale: 0 = no decay (excellent quality); 1 = slight decay; 2 ≤ 1/4 decay; 3 = 1/4–1/2 decay; 4 ≥ 1/2 decay. The decay rate was calculated as ∑ (decay scale × number of corresponding fruit)/(4 × total number of fruit). 

### 4.3. Determination of Physiological Characters

Respiratory rate was measured by using the infrared gas analyzer (Li-6262 CO_2_/H_2_O analyzer, LI-COR, Inc., USA) according to the method described by Wu et al. [5]. Twelve fruit of each replicate was randomly selected, weighted and placed in a 9.5 L plastic container. The change of CO_2_ concentration was determined and the respiratory rate was expressed as CO_2_ (mg kg^−1^ h^−1^, fresh weight). TSS and TA were determined by using a hand-held digital refractometer (Model PAL-BX|ACID F5, Atago, Tokyo, Japan) according to the manufacturer’s instructions. TSS was expressed as Brix (%). A total of five biological replicates of 0, 1, 4, and 7 d samples were used to the measure respiratory rate, TSS and TA. 

### 4.4. Determination of ROS-Related Characters 

A total of four biological replicates of 0, 1, 4, and 7 d samples were used to the measure ROS-related characters. The content of hydrogen peroxide of pitaya peel was measured according to the method described by Zhang et al. [49]. A 3 g frozen peel powder was used for hydrogen peroxide determination. The content of hydroxyl radical was measured by using the Hydroxyl Radical Assay Kit (A018; Nanjing Jiancheng Bioengineering Institute, Nanjing, China) according to the manufacturer’s instruction [5]. Briefly, a 1 g frozen sample powder was homogenized in 2 mL ultrapure water, 0.2 mL of supernatant was mixed with 0.6 mL of reaction buffer. Then, 2 mL chromogenic agent was added to terminate the reaction. The absorbance of the supernatant was measured at 550 nm.

The activities of peroxidase (POD), ascorbate peroxidase (APX), superoxide dismutase (SOD) and catalase (CAT) were determined according to the method described by Zhang et al. [49]. Peel tissues (2.0 g) were ground and homogenized with 10 mL of 0.2 M sodium phosphate buffer (pH 7.0). The supernatant was collected for enzyme assays. POD activity was measured by incubating 0.05 mL of enzyme extraction and 2.95 mL reaction mixture (0.1 mL of 4.0% guaiacol, 0.1 mL of 0.46% H_2_O_2_ and 2.75 mL of 0.05 M sodium phosphate buffer pH 7.0). One unit of POD activity was defined as the amount of enzyme that caused a 0.01 increase in absorbance at 470 nm per minute. SOD activity was determined by measuring its ability to inhibit the photoreduction of nitro-blue-tetrazolium (NBT). One unit (U) of SOD activity is defined as the amount of enzyme that causes a 50% inhibition of NBT reduction at 560 nm. CAT activity was measured by monitoring the decomposition of H_2_O_2_ at 240 nm. One unit (U) of CAT activity is defined as the amount of enzyme that decomposes 1 nmol of H_2_O_2_ per minute. The APX activity was determined by monitoring the oxidation of ascorbic acid in the presence of H_2_O_2_. One unit (U) of APX activity is defined as the amount of enzyme that oxidizes 1 μmol ascorbate per minute.

The peel tissues (2.0 g) were ground and then extracted with 20 mL methanol for 30 min. After centrifugation at 15,000 *g* for 20 min, the supernatants was collected for the determination of DPPH radical scavenging activity and reducing power according to the method described by Duan et al. [8]. Briefly, 0.1 mL of supernatant with 2.9 mL of 0.1 mM DPPH dissolved in methanol solution, and the absorbance was measured at 517 nm to calculate the DPPH radical scavenging activity. For reducing power, 0.1 mL of supernatant was mixed with 2.5 mL of 0.2 mM phosphate buffer (pH 6.6) and 2.5 mL of 1% potassium ferricyanide. After incubation at 50 °C for 20 min, 2.5 mL of 10% trichloroacetic acid was added, then 5 mL of distilled water and 1 mL of 0.1% ferric chloride were added. The absorbance was measured at 700 nm. 

### 4.5. Primary Metabolite Profiling of Pitaya Peel

Primary metabolomics profiling was performed based on the method described by Yun et al. [50] with some modifications. A 200 mg sample was added to 1800 µL methanol (−20 °C) for extraction. A 200 µL of 0.2 mg mL^-1^ ribitol in water was added as an internal standard for quantification. The extract was incubated with ultrasonic treatments at 4 °C and then heated in a water bath at 70 °C for 15 min. The extract was put into the refrigerator at −20 °C for 0.5 h, and then centrifuged at 5000 *g* at 4 °C for 15 min. A 100 µL supernatant was transferred to a glass vial and dried in a rotary vacuum evaporator at 30 °C. Dry precipitate was dissolved in 80 µL of 20 mg mL^-1^ methoxyamine hydrochloride in pyridine and incubated in the oven for 1.5 h at 37 °C, then 80 µL of MSTFA (N-Methyl-N-(trimethylsilyl) trifluoroacetamide) was added and incubated at 37 °C for 0.5 h. The derivative solution was filtered through a 0.22 µm microporous membrane. A total of four biological replicates of 1 and 7 d samples were used to analyze the primary metabolomics profiling.

### 4.6. Volatile Aroma Analysis

Volatile compounds were determined using SPME coupled with GC-MS as described by Wu et al. [6]. Briefly, a 4 g of fruit peel was suspended in 4 mL saturated sodium chloride solution in a glass vial. Cyclohexanone was added as an internal standard for quantification. A total of four biological replicates of 1 and 7 d samples were used to determine the volatile compounds.

### 4.7. Metabolites Identification

Both primary metabolites and volatile compounds were identified by searching in the NIST05 and NIST database as described by Wu et al. [6]. The search results were further confirmed according to the retention time, similarity index and retention index as described in previous studies [5,50]. Metabolites with similarity index greater than 80 can be used for further analysis in this study. In addition, many primary metabolites and volatile compounds have also been verified by using external standard, mainly including D-glucose, D-fructose, D-arabinose, succinic acid, malic acid, L-serine, L-proline, L-alanine, inositol, hexanal, 2-octenal, 1-hexanol, longifolene, methyl octylate, dodecane, dodecane, hexadecane, pentadecane, heptadecane, and nonadecane. 

### 4.8. Statistical Analysis

Data presented in this study were the mean values of four biological replicates. The significance of difference was calculated by one-way analysis of variance (ANOVA) using SPSS version 16.0 and the significant differences were determined by the Duncan’s test (*p* < 0.05). The data of metabolites was also analyzed for PLS-DA and OPLS-DA by using SIMCA software (Version 15.0, Umetrics, Umea, Sweden). PLS-DA was used for comparative analysis of multiple samples. In this study, peel samples were used as observations and metabolites as variables. In the PLS-DA of primary metabolites, it contained sixteen observations and fifty-five variables. In the PLS-DA of volatile compounds, it contained sixteen observations and forty-nine variables. There were fewer variables associated with the observations, which might lead to some deviations in the data calculation. Therefore, OPLS-DA was used to analyze the key metabolites between two samples. VIP was used in key metabolites analysis in OPLS-DA. In this study, the key metabolites with *p* < 0.05 and VIP > 1 have been paid more attention.5. 

## 5. Conclusions

During the postharvest storage of pitaya fruit, H_2_O_2_ mediated a series of physiological changes, which led to fruit senescence and decay. In this process, TSS and TSS/TA ratio decreased, TA, and respiratory rate increased, and cell wall degraded (Figure 8). Red light irradiation enhanced glycolysis, TCA cycle, aldehydes metabolism, and antioxidant enzymes activities at the early stage of postharvest storage, leading to the reduction of H_2_O_2_, soluble sugars, organic acids, and C-6 and C-7 aldehydes (Figure 8). Therefore, the fruit resistance was significantly induced by red light irradiation at the early stage of postharvest storage. At the later stage of postharvest storage, a larger number of resistance-related metabolites and enzyme activities were induced after red light irradiation, such as SOD, APX, DPPH radical-scavenging, reducing power, fatty acids, and volatile aroma (Figure 8). Those metabolites and enzyme activities might also play an important role in red light irradiation delaying the senescence and decay of pitaya fruit.

## Figures and Tables

**Figure 1 metabolites-10-00108-f001:**
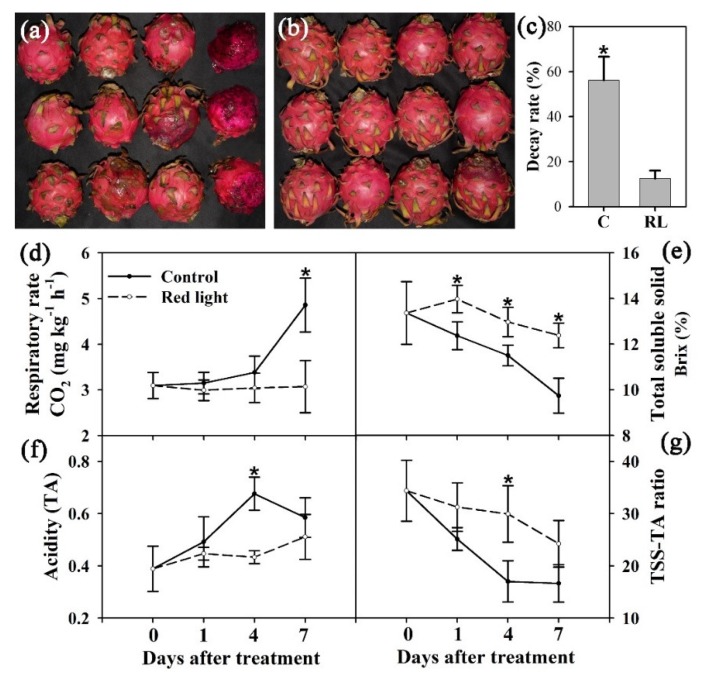
The effects of red light on pitaya fruit. The decay rate of pitaya fruit was measured on Day 10; (**a**): control. (**b**): red light. (**c**): decay rate. Control fruit: C; red light-treated fruit: RL. (**d**): respiratory rate. (**e**): total soluble solid. (**f**): acidity. (**g**): ratio of total soluble solid/acidity. The * on the error bar indicated statistically significant differences (*p* < 0.05) between control and red light-treated samples.

**Figure 2 metabolites-10-00108-f002:**
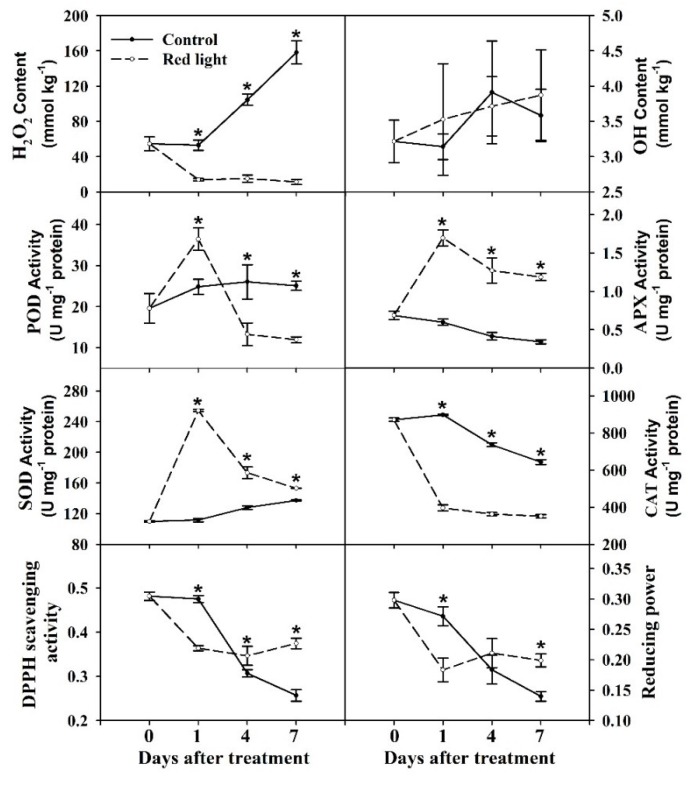
Reactive oxygen species (ROS)-related characters changes after red light irradiation. Mean values and standard errors of four biological replicates were used to draw this graph. The * on the error bars indicated statistically significant differences (*p* < 0.05) between control and red light-treated samples.

**Figure 3 metabolites-10-00108-f003:**
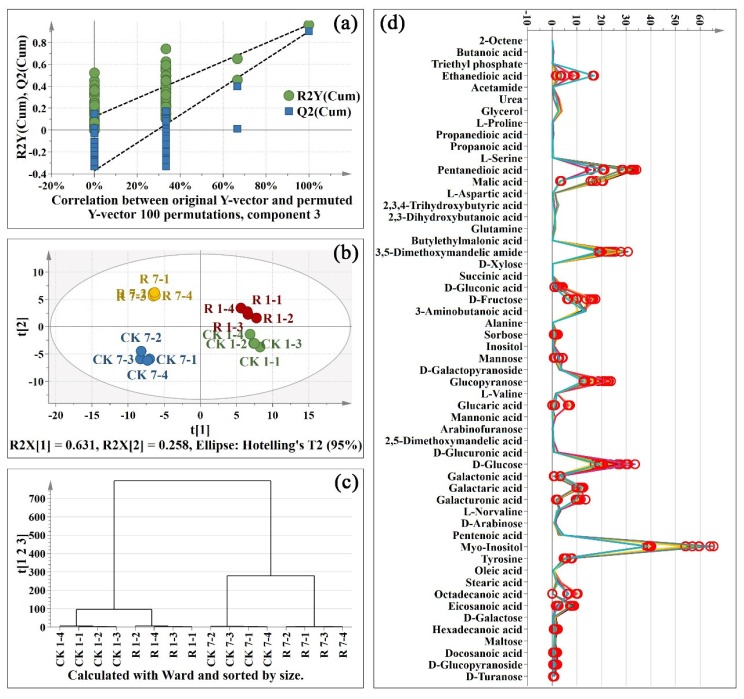
Projection to latent structures discriminant analysis (PLS-DA) of primary metabolites; (**a**): permutations plot of PLS-DA; (**b**): score scatter plot of PLS-DA; (**c**): hierarchical clustering analysis (HCA) plot of PLS-DA; (**d**): spectra plot of PLS-DA. CK: control sample. R: red light. 1-1: one replicate of 1 d sample; 7–2: one replicate of 7 d sample. There are four replicates per sample. The key metabolites in orthogonal projection to latent structures discriminant analysis (OPLS-DA) were marked in red circles in Figure 3d.

**Figure 4 metabolites-10-00108-f004:**
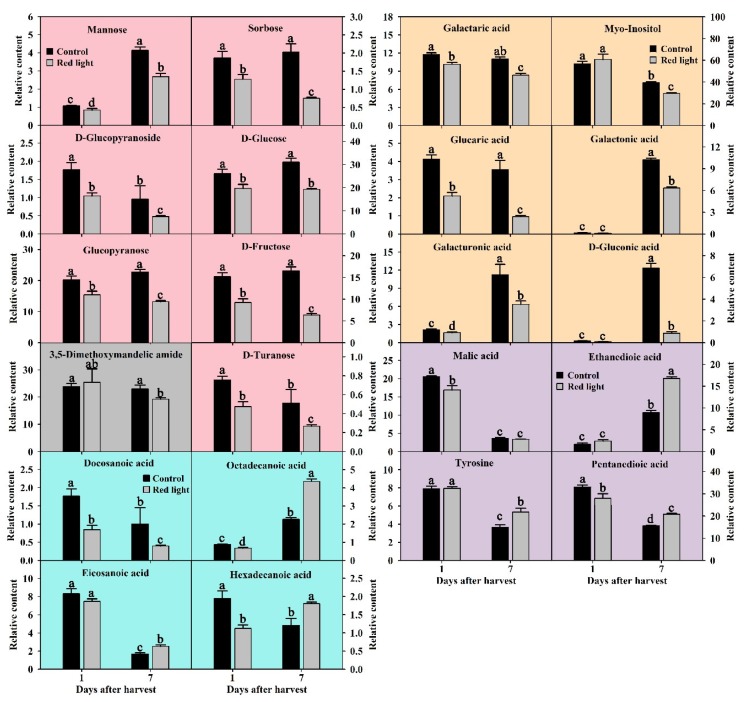
The changes of key primary metabolites after red light irradiation. Soluble sugars were shown in red-pink bar charts. Cell wall degraded products were shown in yellow bar charts. Fatty acids were shown in blue bar charts. Organic acids and amino acid were shown in purple bar charts. Others were shown in gray bar charts. The significant differences were labelled with letters a, b, c, and d, according to Duncan’s test (*p* < 0.05).

**Figure 5 metabolites-10-00108-f005:**
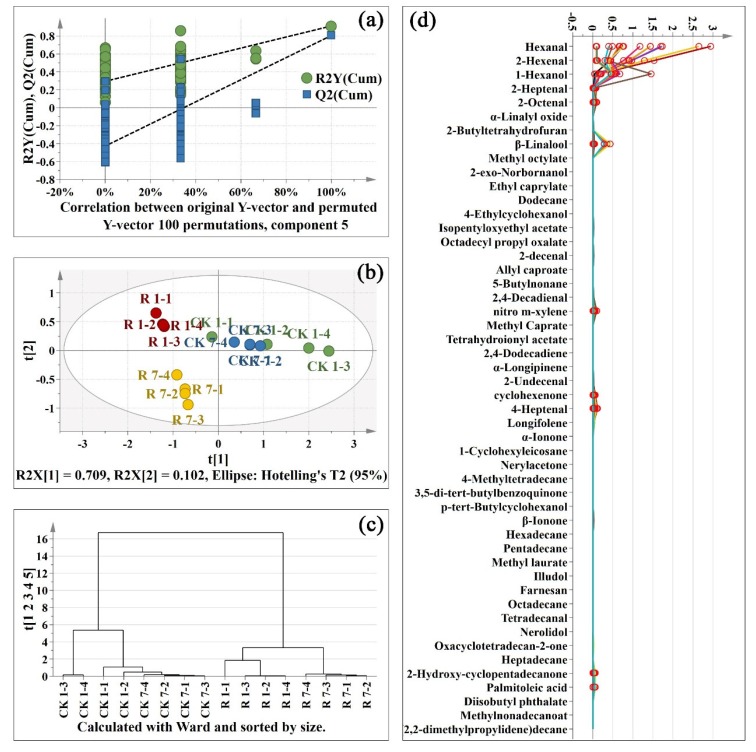
PLS-DA of volatile compounds after red light irradiation; (**a**): permutations plot of PLS-DA; (**b**): score scatter plot of PLS-DA. (**c**): HAC plot of PLS-DA; (**d**): spectra plot of PLS-DA. CK: control sample. R: red light. 1-1: one replicate of 1 d sample; 7–2: one replicate of 7 d sample. There are four replicates per sample. The key volatiles in OPLS-DA were marked in red circles in Figure 5d.

**Figure 6 metabolites-10-00108-f006:**
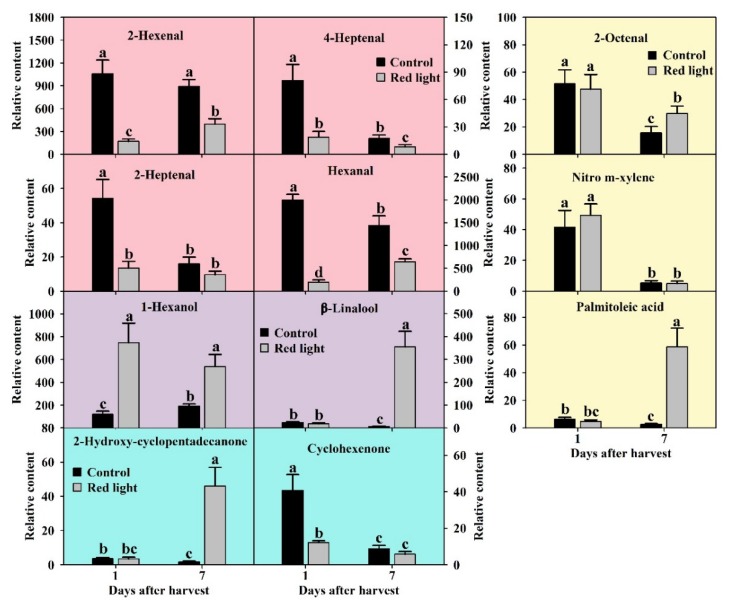
The changes of key volatile compounds after red light irradiation. Aldehydes were shown in red-pink bar charts. Volatile alcohols were shown in purple bar charts. Ketones were shown in blue bar charts. The others were shown in yellow bar charts. The significant differences were labeled with letters a, b and c according Duncan’s test (*p* < 0.05).

**Figure 7 metabolites-10-00108-f007:**
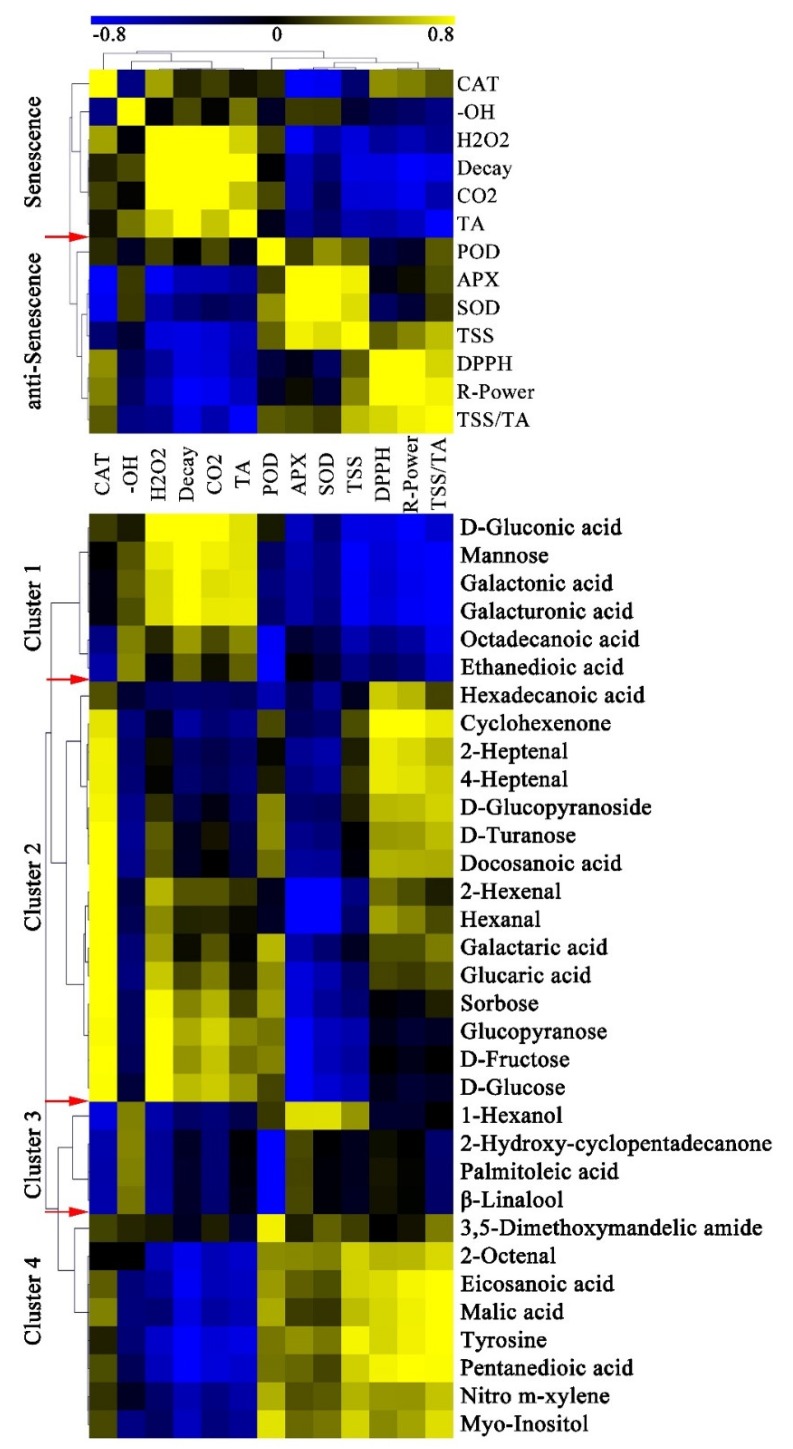
Correlation analysis of physiological characters and key metabolites. The correlation coefficient is represented by virtual color as indicated in the color key. CO_2_: Respiratory rate; DPPH: 1,1-diphenyl-2-picryl-hydrazyl (DPPH) radical-scavenging activity; R-Power: Reducing power. The red arrows indicated the boundaries of clusters.

**Figure 8 metabolites-10-00108-f008:**
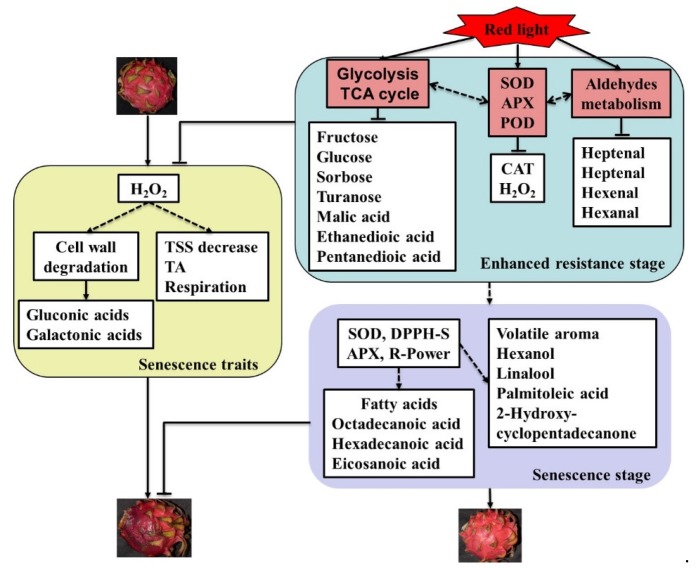
Presumptive model depicting the mechanism in red light irradiation delaying fruit senescence and decay of pitaya. DPPH-S: DPPH radical-scavenging activity; R-Power: Reducing power.

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
