# Peer review of "Deciphering the Metabolic Pathways of Pitaya Peel after Postharvest Red Light Irradiation"

_metabolites, 2020, doi:10.3390/metabo10030108_

Round 1

Reviewer 1 Report

Line 77 the raw  decay data does not support the number of significant figures reported here (e.g. 56.21%). 

Biological sample /replicate versus a replicate?

Figures 4,6 legend needs stat symbol designation so figures stands alone.

Methods:

What criteria were used to evaluate % maturity of the fruit?

Line 477   How was peel obtained?  % of fruit? This detail is needed if an independent investigator were to repeat the study.

Discussion:

Given that one objective of the described study is to evaluate the effect of red light treatment on shelf life, how does red light treatment compare to blue light treatment ( ref 6)?  What information presented in ref.6 lead to the the red light experiments?  Is the peel of this variety photosynthetic at the maturity used?

Author Response

Line 77 the raw decay data does not support the number of significant figures reported here (e.g. 56.21%). 

Response: Figure 1c is drawn using the decay data after calculation. The decay area on each peel of 50 fruit was calculated to evaluate the decay rate of pitaya fruit. Decay scale: 0 = no decay (excellent quality); 1 = slight decay; 2 ≤ 1/4 decay; 3 = 1/4-1/2 decay; 4 ≥ 1/2 decay. The decay rate was calculated as ∑ (decay scale × number of corresponding fruit)/(4 × total number of fruit).

Biological sample /replicate versus a replicate?

Response: Mixed samples were used in this study due to the great difference between single fruit. Twelve pitaya fruit for each biological replicate was sampled, and a total of five biological replicates were collected for each sample. Data presented in this study were the mean values of four biological replicates.

Figures 4,6 legend needs stat symbol designation so figures stands alone.

Response: Yes, thanks. We have added ‘The significant differences were labelled with letters a, b, c and d according Duncan’s test (p<0.05).’ in the legends of Figures 4 and 6.

Methods:

What criteria were used to evaluate % maturity of the fruit?

Response: Pitaya fruit from flowering to ripening only about 40 days. At about 37 days after flowering, the sweetness of pulp increased significantly, and the peel color changed from cyan to fuchsia. Since there are great differences among individual fruit, and the maturity of fruit in different regions and years also varies greatly, we generally define the maturity of fruit as 80-90%.

Line 477   How was peel obtained?  % of fruit? This detail is needed if an independent investigator were to repeat the study.

Response: Thanks for your valuable comment. The middle part of the fruit peel was used for sampling. If a portion of the peel had decayed, the un-decayed portion was sampled. See Lines 478-479 for details.

Discussion:

Given that one objective of the described study is to evaluate the effect of red light treatment on shelf life, how does red light treatment compare to blue light treatment ( ref 6)?  What information presented in ref.6 lead to the red light experiments?  Is the peel of this variety photosynthetic at the maturity used?

Response: Yes, thanks for your valuable comment. Red light treatment and blue light treatment are two independent experiments. We have added comparison between different light irradiations in the Discussion. Detail information was shown in Lines 347-348, 372-374, 403-405, 455-457. As for whether the pitaya fruit still carries out photosynthesis after harvest, we have not done relevant experiments. Since the peel of pitaya contains chlorophyll, we speculated that it might undergo photosynthesis, but the intensity might not be high.

Reviewer 2 Report

Overall, this is a very well-written paper and contains some interesting findings regarding shelf life in pitaya. The analytical chemistry methods employed appear not only reasonable but well-described and replicable. The results and discussion are very clear. The only true concern I had, above minor wording corrections, lays in the limitations of the multivariate statistics employed given the sample size, but I think the authors could simply address this by directly stating the limitations in the manuscript without making too many changes. Overall, very interesting and well-prepared work.

Abstract

Lines 18-19: While not incorrect, you can simply state gas chromatography-mass spectrometry, as it is understood that these two instrumental components are coupled to each other.

Introduction

Line 35: Missing the before pitaya.

Line 40: Those should be these.

Lines 40-41: While it is true that many of these treatments are chemical treatments, that doesn’t necessarily mean that they have a negative impact on human health. They may actually have a positive impact, in some regards. The bigger concern is the additional processing costs that are incurred during the production of extracts or the general use of chemical treatments.

Results

Lines 128-139: Please define the acronyms used. For example, in line 28, you should first state partial least squares-discriminant analysis and then follow that with the PLS-DA acronym.

General: One of the issues that you should probably state in both the Methods and the Results is that the multivariate procedures you employed do have limitations when you are dealing with this small of a number of observations. I agree that PLS-DA is the best way to go about this and it is perfect for instances where you have more variables than observations, but you only have 2 treatments x 4 times x 4 reps for most of your analyses, leaving you with only 32 observations total, and if you are working with the means during this, 8 observations. The hierarchical clustering methods really should have 10 times as many observations as there are variables. It doesn’t mean that these multivariate techniques are not sometimes employed in the literature when there are few observations, but there are very few observations relative to the number of variables in this study, posing some limitations that should be acknowledged. On the bright note, this is potentially foundational for future research in this area, to verify that what you saw looking at a very small dataset is true when you look at a more complete dataset.

In a few instances (e.g. line 136), analysis should be analyze or vice versa.

Line 309: Delete the before cluster.

Consider capitalizing the word cluster as it relates to individual clusters. For example, in line 309, you could write, “In Cluster 2…”.

Discussion

The first paragraph of the discussion section needs some work. The first two sentences need to be formalized a bit (for instance, “leads a lot” in sentence 2). Additionally, I believe I know where the authors were heading with the discussion regarding the pathogens, but I believe there is a critical connecting sentence that is missing that makes this a bit more confusing. Please consider adding a sentence that states multiple fungal pathogens reside on the surface of the fruit and are able to enter the fruit following senescence and the weakening of the peel.

Methods

Line 461: It took me a bit of flipping back and forth to see why there were four sampling times for some analyses and five for another. I also saw four and five biological replicates stated in different places. Of course, the time point does not constitute a biological replicate, so please make sure to make this distinction clear. It might be worth stating earlier in the paper exactly how many time points and exactly how many biological replicates there were for each portion of the study.

Lines 557-565: Same critique as above.

Author Response

Overall, this is a very well-written paper and contains some interesting findings regarding shelf life in pitaya. The analytical chemistry methods employed appear not only reasonable but well-described and replicable. The results and discussion are very clear. The only true concern I had, above minor wording corrections, lays in the limitations of the multivariate statistics employed given the sample size, but I think the authors could simply address this by directly stating the limitations in the manuscript without making too many changes. Overall, very interesting and well-prepared work.

Response: Thanks for your positive comment. We will carefully revise the manuscript according to your comments so as to meet the publication requirements of Metabolites.

Abstract

Lines 18-19: While not incorrect, you can simply state gas chromatography-mass spectrometry, as it is understood that these two instrumental components are coupled to each other.

Response: Thanks for your comment. We have changed it into ‘gas chromatography mass spectrometry (GC-MS)’.

Introduction

Line 35: Missing the before pitaya.

Response: Yes, thanks. We have revised it accordingly.

Line 40: Those should be these.

Response: Yes, thanks. We have revised it accordingly.

Lines 40-41: While it is true that many of these treatments are chemical treatments, that doesn’t necessarily mean that they have a negative impact on human health. They may actually have a positive impact, in some regards. The bigger concern is the additional processing costs that are incurred during the production of extracts or the general use of chemical treatments.

Response: Thanks for your valuable suggestion. We have added a sentence in Lines 40-43. In addition, the production cost of extract or the processing cost of chemical treatment is also an important aspect that affects its application promotion. Therefore, the developing green, safe, and low-cost physical preservation treatment has high application value.

Results

Lines 128-139: Please define the acronyms used. For example, in line 28, you should first state partial least squares-discriminant analysis and then follow that with the PLS-DA acronym.

Response: Yes, thanks for your comment. We have revised it accordingly. PLS-DA: projection to latent structures discriminant analysis; OPLS-DA: orthogonal projection to latent structures discriminant analysis; VIP: variable importance in projection. See Lines 130-142 for details.

General: One of the issues that you should probably state in both the Methods and the Results is that the multivariate procedures you employed do have limitations when you are dealing with this small of a number of observations. I agree that PLS-DA is the best way to go about this and it is perfect for instances where you have more variables than observations, but you only have 2 treatments x 4 times x 4 reps for most of your analyses, leaving you with only 32 observations total, and if you are working with the means during this, 8 observations. The hierarchical clustering methods really should have 10 times as many observations as there are variables. It doesn’t mean that these multivariate techniques are not sometimes employed in the literature when there are few observations, but there are very few observations relative to the number of variables in this study, posing some limitations that should be acknowledged. On the bright note, this is potentially foundational for future research in this area, to verify that what you saw looking at a very small dataset is true when you look at a more complete dataset.

Response: Thanks for your valuable comment. We have noted the limitations in Lines 586-592. In this study, peel samples were used as observations and metabolites as variables. In the PLS-DA of primary metabolites, it contained sixteen observations and fifty-five variables. In the PLS-DA of volatile compounds, it contained sixteen observations and forty-nine variables. There were fewer variables associated with the observations, which might lead to some deviations in the data calculation. Therefore, OPLS-DA was used to analyze the key metabolites between two samples in our study.

In a few instances (e.g. line 136), analysis should be analyze or vice versa.

Response: Yes, we have revised it accordingly in Lines 133, 141 and 232.

Line 309: Delete the before cluster.

Response: Thanks, we have deleted ‘the’ in Lines 306, 308, 315, 319 and 324.

Consider capitalizing the word cluster as it relates to individual clusters. For example, in line 309, you could write, “In Cluster 2…”.

Response: Yes, thanks. We have revised it accordingly. See Lines 308-324 for details.

Discussion

The first paragraph of the discussion section needs some work. The first two sentences need to be formalized a bit (for instance, “leads a lot” in sentence 2). Additionally, I believe I know where the authors were heading with the discussion regarding the pathogens, but I believe there is a critical connecting sentence that is missing that makes this a bit more confusing. Please consider adding a sentence that states multiple fungal pathogens reside on the surface of the fruit and are able to enter the fruit following senescence and the weakening of the peel.

Response: Thanks for your valuable comment. We have revised the sentence 2 into ‘During the senescence, the fruit gradually deteriorated and disordered’. In addition, we have added a sentence between senescence and pathogens. See Lines 334-341 for details.

Methods

Line 461: It took me a bit of flipping back and forth to see why there were four sampling times for some analyses and five for another. I also saw four and five biological replicates stated in different places. Of course, the time point does not constitute a biological replicate, so please make sure to make this distinction clear. It might be worth stating earlier in the paper exactly how many time points and exactly how many biological replicates there were for each portion of the study.

Lines 557-565: Same critique as above.

Response: I'm sorry for the confusion caused to you by the unclear description. We have added a sentence in each method to clarify the usage of time points and biological replicates. Details were shown in Lines 494-495, 497-498, 539-540, 559-560 and 572-579.

Reviewer 3 Report

This manuscript explores the potential of red light irradiation (RLI) as a suitable treatment to delay senescence of pitaya fruit. The Authors report a considerable amount of work and, albeit showing what in my view represent some minor flaws,

Author Response

The Authors explore the potential of red light irradiation (RLI) as a treatment to delay senescence of pitaya fruit. The Authors have undertaken a wide range of biochemical analysis resulting in a large amount of valuable data. In my opinion, however, the manuscript suffers from some flaws that should be corrected before being considered suitable for publication. My comments are as follows:

Response: Thanks for your positive comments.

  1. The use of the English language requires some attention. Although generally understandable throughout the text, there is ample opportunity for grammatical improvements. This might sound as a “default” remark, but it is not.

Response: Yes, we have revised the grammar carefully.

  1. A number of format inconsistencies and mistakes can be found in the reference list. Additionally, please check the correctness for journal name abbreviations (for example, references 13, 19, 25, 26, 46, 47). Journal name for reference 16 is missing. Page information is incomplete for reference 21. Please revise the whole section.

Response: Yes, thanks for your comment. We have revised the whole ‘references’ section accordingly. See the ‘references’ for details.

  1. Check caption to Figure 2: the Authors are NOT showing bars.

Response: Yes, they are error bars. We have revised it.

  1. I personally would rather have Figures 4 and 6 transformed into Tables, for the sake of clarity and reader-friendliness.

Response: Figures 4 and 6 depict our selected key metabolites, and we feel that the use of the pictorial form can better highlight the effect of red light treatment on the relative content of these metabolites.

In addition to these “formal” flaws, I have also some remarks on the contents of the manuscript, namely:

  1. Line 29: I am surprised that the Authors state that pitaya is a “new” fruit crop. I infer that they mean rather that pitaya production is increasing outside the traditional growing areas. I strongly suggest to reword this sentence.

Response: Yes, thanks. We have revised it into ‘Pitaya (Hylocereus polyrhizus) is a fruit crop of Latin America origin, and red pitaya is now widely cultivated in many other tropical and subtropical regions due to its attractive pulp color and high nutritional benefits’.

  1. In my view, an important concern with this work is, how all the reported biochemical changes translated into ACTUAL fruit quality. Certainly, the Authors assessed a few physiological and eating quality-related traits (Figure 1), and it seems clear that decay rates 10 days after harvest were much lower in treated fruit. However, the Authors themselves state that the usual shelf life of pitaya fruit after harvest is around 5 days (line 34). Therefore, I wonder whether treatment benefits were actually perceivable by potential consumers, which is a relevant question when considering the implementation of a given postharvest procedure. Indeed, it might happen that the cost of treatment application outweighs the benefits. The Authors overlooked some other attributes that could have helped in data interpretation. For example, in section 2.3.2 changes in “cell wall degradation” products are presented and it is suggested that cell wall disassembly was delayed by RLI. Yet did these changes translate in perceivably delayed firmness loss? No firmness levels are reported, which is a quite “core” determination when assessing fruit quality. Similarly, section 2.3.3 describes changes in fatty acids and highlights the fate of very-long-chain fatty acids (VLCFA) at both the “enhanced resistance” and the senescence stages. This led me to think these changes might have something to do with cuticle integrity and functions. It is a big pity that weight loss of fruit was not determined; these data are very simple to obtain and would have provided useful information. Without negating the interest of data reported in the manuscript, I suggest these aspects be taken in consideration when discussing results and the potential of commercial application of RLI.

Response: Red light treatment has been widely used in the fruit development stage of pitaya in the cultivation process, and red light treatment can indeed prolong the shelf life of pitaya fruit. Since its processing cost has not been accounted for, there is no data base on the comparison between its applied economic value and its costs. Due to the long processing time of red light, there is no postharvest equipment using red light commercially. As for the red light treatment effect, it makes more sense for pitaya fruit sellers. Due to the limitations of the experimental design at the time, many meaningful data for evaluating fruit quality were not measured before sampling, such as firmness, weight loss, cuticle integrity, and others. We have discussed the application value of red light in postharvest treatment. See Lines 345-347 for details.

  1. How would the Authors reconcile a delaying effect of RLI on fruit senescence with results suggesting that the treatment accelerated the generation of ATP? (acceleration of glycolysis and Krebs cycle?).

Response: Red light treatment accelerates the consumption of soluble sugars and organic acids, which are mostly substrates of glycolysis and TCA cycle, so we speculate that red light treatment may accelerate the glycolysis and TCA cycle of pitaya peel. This process mainly occurs during red light processing (1 d), and we speculate that the generated ATP may be used in induced resistance of pitaya peel.

  1. Lines 412-415: Please note that nothing of the like can be inferred without sensory analyses having been performed! (please see previous point 6). The aroma profile of a fruit is complex, and many interactions exist among the different compounds present; it is NOT a simple question of the amount of emitted compounds. It is a matter of concentration, but also of odour thresholds, odour description, and of balance and interactions among compounds. Which redirects me to the same question as above: were all these changes actually perceivable by potential consumers?
  2. In connection with above, please note that the reported decrease in aldehyde concentration may have actually translated into limited supply of precursors for ester biosynthesis, and thus have been detrimental rather than beneficial for fruit aroma. Once more, mere biochemical data mean little in the absence of eating and sensory quality data.

Response: Take comments 8 and 9 together. Yes, thanks for your valuable comment. We have deleted ‘red light make the aroma of pitaya peel more attractive’. See Lines 426-428 for details.

  1. Materials and methods, section 4.1: Please provide information on which indices were used to determine “commercial maturity stage”.

Response: Pitaya fruit from flowering to ripening only about 40 days. At about 37 days after flowering, the sweetness of pulp increased significantly, and the peel color changed from cyan to fuchsia. Since there are great differences among individual fruit, and the maturity of fruit in different regions and years also varies greatly, we generally define the maturity of fruit as 80-90%. We have revised it accordingly. See Lines 467-468 for details.

  1. Materials and methods, section 4.3: As to fruit used for the determination of respiratory rates, were they placed continuously (i.e. throughout the whole 10 days experimental period) in the plastic container?

Response: The fruit was selected at random, and used to measure physiological indicators such as TTS and TA immediately after the measurement of respiratory rate. We have revised it into ‘Twelve fruit of each replicate was randomly selected, weighted and placed in a 9.5 L plastic container’.

Reviewer 4 Report

Dear authors,

The paper entitled "Deciphering the metabolic pathways of pitaya peel after postharvst red light irradiation" is a well-written paper with a large number of experiements and results. However the scientific approach is not original, similar material and methods have been already described with ozone treament and blue light treatment. 

Can the authors explain their choice for the red light irradiation at 660 nm, the irradiation timing (24h 100 LUX) and the distance (40 cm) from the leds to the fruits?

The authors show that the red light irradiation delays the senescence and the decay of pitaya fruits. Under blue light, authors also showed a delay in the decay of pitaya fruits as well as in the decline of the quality of fruits. What is the true difference between light irradiations?

Even if the paper present a lot of interesting results, the authors must construct their results and their discussion by the comparison between different light irradiations.

For that reason I do not accept the paper for publication.

Author Response

The paper entitled "Deciphering the metabolic pathways of pitaya peel after postharvst red light irradiation" is a well-written paper with a large number of experiments and results. However the scientific approach is not original, similar material and methods have been already described with ozone treatment and blue light treatment.

Response: Yes, similar materials and methods have been described in previous experiments with ozone treatment and blue light treatment. However, many ROS related characters and metabolites differ significantly in response to different treatments. For example, red light treatment and ozone treatment significantly decreased the content of H2O2 on day 1, while blue light treatment significantly increased the content of H2O2 on day 1. Blue light treatment and ozone treatment significantly decreased the content of -OH on day 7, while the content of -OH showed slight change after red light treatment on day 7. On day 1, most of key soluble sugars were induced by blue light treatment but consumed by the red light treatment. There are many other examples of metabolites changing differently after different treatments. It indicated that different treatments had different effects on the metabolic pathways of pitaya fruit.

Can the authors explain their choice for the red light irradiation at 660 nm, the irradiation timing (24 h 100 LUX) and the distance (40 cm) from the leds to the fruits?

Response: The duration of illumination was screened in previous trials. The wavelength, intensity, and distance of LED light from the fruit were matched according to the reported references.

Zhang, N., et al. (2016). "Effects of Red Light-Emitting Diode (LED) on the Postharvest Yellowing Change of Broccoli." Spectroscopy and Spectral Analysis 36(4): 955-959.

Ma, G., et al. (2012). "Effect of Blue and Red LED Light Irradiation on beta-Cryptoxanthin Accumulation in the Flavedo of Citrus Fruits." Journal of Agricultural and Food Chemistry 60(1): 197-201.

Yamaga, I., et al. (2016). "Rind Color Development in Satsuma Mandarin Fruits Treated by Low-intensity Red Light-emitting Diode (LED) Irradiation." Food Science and Technology Research 22(1): 59-64.

The authors show that the red light irradiation delays the senescence and the decay of pitaya fruits. Under blue light, authors also showed a delay in the decay of pitaya fruits as well as in the decline of the quality of fruits. What is the true difference between light irradiations?

Response: Yes, both blue light and red light irradiation could significantly delay the senescence and the decay of pitaya fruit. However, many ROS related characters and metabolites of pitaya peel differ significantly in response to different treatments, indicating that different treatments had different effects on the metabolic pathways of pitaya fruit.

Even if the paper present a lot of interesting results, the authors must construct their results and their discussion by the comparison between different light irradiations.

Response: Yes, thanks for your valuable comment. We have added comparison between different light irradiations in the Discussion. Detail information was shown in Lines 347-348, 372-374, 403-405, 455-457.

For that reason I do not accept the paper for publication.

Response: I am sorry that we have not made a comparative analysis of blue light treatment and red light treatment. Although blue and red light can significantly delay the senescence and decay of pitaya fruit, there are significant differences in their metabolism regulation, such as ROS, soluble sugars, fatty acids and some volatile aromas. We added the comparison of different light irradiation in the Discussion, hoping to meet your requirements.

Round 2

Reviewer 1 Report

Methods: The authors have not adequately described method of attaining the peel. If someone handed me a pitaya fruit how would I collect the peel? I do not find in the revised manuscript a means of estimating the 80-90% mentioned. Was surface color measured/estimated, soluble solids?

The point: how could I repeat your experiments?

Results: The authors should review significant figures. Numbers derived by multiplication/division should have the same relative uncertainty as the raw data. Your relative uncertainty in raw data is 1 in 5 at best which does not support the 1 in 5000 noted in the original.

Author Response

Methods: The authors have not adequately described method of attaining the peel. If someone handed me a pitaya fruit how would I collect the peel? I do not find in the revised manuscript a means of estimating the 80-90% mentioned. Was surface color measured/estimated, soluble solids?

Response: We apologize for not describing the mature stage and sampling details of the fruit. We have revised it accordingly. Please see Lines 467-478 for details. Generally, we define the maturity of fruit as 80-90%, the sweetness of pulp increased significantly, and the peel color changed from cyan to fuchsia. If the harvest period continuous rainfall, fruit color and sweetness are relatively low. Since there are great differences among individual fruit, and the maturity of fruit in different regions and years also varies greatly, fruit maturity cannot be determined by the value of peel color or TSS.

The point: how could I repeat your experiments?

Response: I'm sorry for the confusion caused to you because the material sampling is not clearly described. We have revised it accordingly. Please see Lines 467-478 for details.

Results: The authors should review significant figures. Numbers derived by multiplication/division should have the same relative uncertainty as the raw data. Your relative uncertainty in raw data is 1 in 5 at best which does not support the 1 in 5000 noted in the original.

Response: Sorry, we don't quite understand what you describes. In this study, many data are based on specific experimental methods, only the decay rate and TSS-TA ratio were calculated by using the multiplication and division. The standard error is calculated by repeated experiments. Do you mean the standard errors of TSS-TA ratio and decay rate are too low?

For the TSS-TA ratio, this is a mature algorithm, has been applied to many fruit quality analysis. I will not give an example here.

For the decay rate, because the number of fruit used in the calculation is high, the experimental error is small. This method is also used to calculate the Browning of litchi fruit. Such as:

Zhang, Z. K., et al. (2015). "Enzymatic browning and antioxidant activities in harvested litchi fruit as influenced by apple polyphenols." Food Chemistry 171: 191-199.

Yun, Z., et al. (2016). "Comparative transcriptome and metabolome provides new insights into the regulatory mechanisms of accelerated senescence in litchi fruit after cold storage." Scientific Reports 6: 167.

Zhang, D. D., et al. (2018). "6-Benzylaminopurine improves the quality of harvested litchi fruit." Postharvest Biology and Technology 143: 137-142.

Reviewer 4 Report

The paper entitled "Deciphering the metabolic pathways of pitaya peel after postharvest red light irradiation " has been revised properly. The role of red light  irradiation (this paper) and the role of blue light irradiation (ref 6) are better explained, and the differences they produce in the regulation of ROS signals as well as on fruit resistance are more precise.

I accept the paper for publication.

Author Response

The paper entitled "Deciphering the metabolic pathways of pitaya peel after postharvest red light irradiation" has been revised properly. The role of red light irradiation (this paper) and the role of blue light irradiation (ref 6) are better explained, and the differences they produce in the regulation of ROS signals as well as on fruit resistance are more precise.

I accept the paper for publication.

Response: Thanks for your positive comment.